# Implication of Cellular Senescence in Osteoarthritis: A Study on Equine Synovial Fluid Mesenchymal Stromal Cells

**DOI:** 10.3390/ijms24043109

**Published:** 2023-02-04

**Authors:** Gabriella Teti, Eleonora Mazzotti, Valentina Gatta, Francesca Chiarini, Maria Laura Alfieri, Mirella Falconi

**Affiliations:** 1Department of Biomedical and Neuromotor Sciences, University di Bologna, 40126 Bologna, Italy; 2Boehringer Ingelheim Vetmedica GmbH, 55218 Ingelheim am Rhein, Germany; 3Department of Biomedical, Metabolic and Neural Sciences, University of Modena and Reggio Emilia, 41125 Modena, Italy; 4Department of Medical and Surgical Sciences, University of Bologna, 40126 Bologna, Italy

**Keywords:** cellular senescence, osteoarthritis, mesenchymal stromal cells, aging

## Abstract

Osteoarthritis (OA) is described as a chronic degenerative disease characterized by the loss of articular cartilage. Senescence is a natural cellular response to stressors. Beneficial in certain conditions, the accumulation of senescent cells has been implicated in the pathophysiology of many diseases associated with aging. Recently, it has been demonstrated that mesenchymal stem/stromal cells isolated from OA patients contain many senescent cells that inhibit cartilage regeneration. However, the link between cellular senescence in MSCs and OA progression is still debated. In this study, we aim to characterize and compare synovial fluid MSCs (sf-MSCs), isolated from OA joints, with healthy sf-MSCs, investigating the senescence hallmarks and how this state could affect cartilage repair. Sf-MSCs were isolated from tibiotarsal joints of healthy and diseased horses with an established diagnosis of OA with an age ranging from 8 to 14 years. Cells were cultured in vitro and characterized for cell proliferation assay, cell cycle analysis, ROS detection assay, ultrastructure analysis, and the expression of senescent markers. To evaluate the influence of senescence on chondrogenic differentiation, OA sf-MSCs were stimulated in vitro for up to 21 days with chondrogenic factors, and the expression of chondrogenic markers was compared with healthy sf-MSCs. Our findings demonstrated the presence of senescent sf-MSCs in OA joints with impaired chondrogenic differentiation abilities, which could have a potential influence on OA progression.

## 1. Introduction

Tissue homeostasis is described as the property of a tissue to maintain its functional capacity during development, adulthood, and aging. This includes the ability of tissue turnover under normal conditions and the ability to regenerate lost tissue upon injury. The equilibrium between the self-repair mechanisms of differentiated cells and their replacement through differentiation of tissue-specific adult stem cells ensures such homeostasis [1]. During aging, we have a progressive loss of this equilibrium due to a time-dependent decrease in stem cell functions, such as self-renewal, differentiation, and tissue repair capacities, which predispose the organism to the development of several age-related diseases [1,2,3]. Recently, it has been proved that tissue accumulation of senescent cells is one of the key processes that contribute to age-related health decline and chronic disease progression [4,5,6].

Cellular senescence is characterized by complex modifications at the cellular and molecular level, which involve changes in the protein expression profile of the cell, leading to replicative arrest and cell dysfunction [7,8,9]. Senescent cells remain viable and show modifications in metabolic activity and resistance to apoptosis. They are also characterized by a strong increase in the secretion of growth factors, inflammatory cytokines, and proteolytic enzymes, termed the “senescence-associated secretory phenotype” (SASP), which is thought to play a key role in tissue dysfunction [4,9]. Although it has beneficial effects such as protection against tumorigenesis, the aberrant accumulation of senescent cells has been associated with several age-related diseases such as osteoporosis, cardiovascular diseases [5], sarcopenia, dementia, pulmonary fibrosis, and osteoarthritis [4].

Osteoarthritis (OA) is the most common degenerative disease in both human and veterinary medicine and is one of the most disabling diseases in developed countries. Global estimates are that 9.6% of men and 18.0% of women > 60 years of age have symptomatic (painful) OA [10]. It is often described as a chronic degenerative disorder and thought by many to be an inevitable consequence of aging. OA is characterized by progressive deterioration and loss of joint cartilage, mainly thought to be induced by wear and tear. During the last decade, however, it became clearer that OA is not purely a mechanical condition but a more complex disease that includes all the joint components [11]. In addition to cartilage degeneration, synovium, subchondral bone, and associated muscles undergo pathological alterations during OA development and progression [10,12]. OA is associated with low-grade systemic and joint inflammation created by proinflammatory and matrix-degrading cytokines [12].

It is clear, however, that the healthy joint requires a fine-tuned balance between molecular signals that regulate homeostasis, damage restoration, and remodeling. This balance is entrusted primarily to the autonomous and non-autonomous functions of the resident mesenchymal stromal cells (MSCs). These non-hematopoietic CD34−, CD105+, CD90+, and CD73+ cells are mainly found in the subchondral bone marrow and the knee synovial tissue [13,14]. Recent studies have proven that synovial fluid mesenchymal stromal cells (sf-MSCs) play a key role in tissue homeostasis [15,16]. It has also demonstrated a great capacity to generate cartilage in both in vivo and in vitro studies [3,17], and they represent one of the best candidates for future cell-based therapies.

During joint aging, the ability of healthy sf-MSCs to differentiate toward a chondrogenic phenotype deeply decreased [3] while the number of senescent cells detected in the synovial fluid and joint tissues increased [12,18], leading to a progressive loss in tissue homeostasis [1,19]. However, senescence seems to be not the only driver of OA, and the connection between cellular senescence and OA onset and development is still poorly described. Moreover, a few experimental data demonstrated a possible connection between cellular senescence in MSCs and OA progression [18]. However, there is a correlation between cellular senescence with age [3]. The chance that this condition occurs in young adults and could be responsible for premature aging which supports the development of age-related disease, is still under investigation.

Therefore, in this study, we aimed to determine the presence and accumulation of senescent sf-MSCs in OA joints of young adult donor and their potential contribution to disease progression. To this purpose, sf-MSCs were isolated from tibiotarsal joints of healthy and diseased horses with an established diagnosis of OA with ages ranging from 3 to 14 years. Due to the high similarity in the anatomy and biomechanics between human and horse knee joints, an equine model was chosen for this study. Sf-MSCs were cultured in vitro and characterized for cell proliferation assay, cell cycle analysis, ROS detection assay, ultrastructure analysis, and evaluation of senescent markers to possibly highlight the presence of a senescence phenotype in cells derived from osteoarthritic joints. To evaluate the influence of OA on chondrogenic differentiation, sf-MSCs isolated from the diseased joint (OA sf-MSCs) were stimulated in vitro for up to 21 days with chondrogenic factors. The expression of chondrogenic markers was compared with healthy sf-MSCs (H sf-MSCs). Our findings showed an accumulation of senescent sf-MSCs in OA joints with impaired chondrogenic differentiation abilities, which could contribute to OA progression.

## 2. Results

### 2.1. sf-MSCs Express Mesenchymal and Self-Renewal Markers

In order to evaluate the ability of sf-MSCs, isolated from healthy and pathological joints, to keep their mesenchymal and self-renewal phenotype, the mRNA expression of the self-renewal markers homeobox protein transcription factor (NANOG), octamer-binding transcription factor 4 (OCT4) and SRY (sex determining region Y)-box 2 (SOX2) genes [20] and of the mesenchymal stromal markers CD90, CD105 were evaluated by RT-PCR. Sf-MSCs, isolated from healthy and OA joints, showed a significant expression of the self-renewal markers NANOG, SOX2, and OCT4 (Figure 1). Furthermore, a positive signal for the mesenchymal markers CD90 and CD105, and a negative signal for the hematopoietic markers CD34, in agreement with the minimum criteria of the International Society of Cellular Therapy [21], was observed in all the samples analyzed (Figure 1). These results demonstrated the ability of both healthy and OA sf-MSCs to keep their mesenchymal and self-renewal phenotype, even in pathological conditions.

### 2.2. sf-MSCs Isolated from OA Joints Are Characterized by a Significant Reduction in the Ability to Proliferate Compared to sf-MSCs Isolated from Healthy Joints

To evaluate possible differences in the proliferation ability of sf-MSCs isolated from OA joints compared with healthy sf-MSCs, a BrdU proliferation assay was carried out. Results showed a significantly reduced proliferation ability of OA sf-MSCs compared with H sf-MSCs (Figure 2a), suggesting a slower capacity of OA sf-MSCs to enter the S phase of the cell cycle compared with cells isolated from healthy joints.

Taking into consideration the results from the proliferation assay, our next step was to investigate differences in the cellular doubling time (DT) in sf-MSCs isolated from pathological and healthy joints. Cells were grown up to 5 passages (p5), and the average DT value for each donor from passage p1 (p1) to p5 was shown in Figure 2b. Not surprisingly, all the pathological donors showed a higher doubling time compared with healthy donors, in agreement with data from cell proliferation assays.

Based on replication assay and duplication time results, we assumed changes in cell cycle progression in healthy and OA sf-MSCs. To this aim, a cell cycle analysis based on PI staining and flow cytometry analysis was carried out. Flow cytometric analysis on PI-stained cells showed a high percentage of healthy sf-MSCs in the G1 phase (average value 72.0% ± 9.2), S phase showed a low percentage (average value 7.8% ± 4.1), while a moderate percentage in G2/M phase was shown (average value 16.1% ± 4.7) (Figure 2c and Table 1). Analyzing the OA group, a reduction of sf-MSCs in the G1 phase (average value 58.8% ± 5.2) was observed in all the donors tested (Figure 2c and Table 1) compared with healthy sf-MSCs, connected with an increase of cells in S phase (average value 16.4 % ± 5.9) and G2/M phase (average value 22.9% ± 7.5) (Figure 2c and Table 1). These results show a significant accumulation of the cells in the G2/M phase with concomitant lag in the cell cycle in OA sf-MSCs compared with healthy ones. Indeed, the percentage of OA sf-MSCs in the S and G2/M phases are higher, suggesting a prolonged time for the cells in these phases.

### 2.3. TEM Analysis Showed a High Activity in Cellular Digestion in H sf-MSCs Compared to OA sf-MSCs

Ultrastructural analysis by TEM was carried out to evaluate morphological intracytoplasmic changes in H and OA sf-MSCs. Figure 3a showed a low-magnification image of healthy sf-MSCs. The cytoplasm is deeply filled by several black round-shaped structures (Figure 3a,b), connected to a high number of primary and secondary lysosomes and products of cellular digestion. Few mitochondria were detected, while a developed Golgi complex and rough endoplasmic reticulum (RER) were observed (Figure 3c). On the contrary and unexpectedly, OA sf-MSCs showed a cytoplasm almost free from lysosomes and end products of cellular digestion (Figure 3d–f), while Golgi complex and RER were well developed and filled most of the cytoplasmic area (Figure 3e,f). Several mitochondria, with different sizes and shapes, were detected (Figure 3e,f).

### 2.4. OA sf-MSCs Showed Higher Intracellular ROS Levels Compared to H sf-MSCs

In order to detect the level of intracellular ROS in OA sf-MSCs, a ROS detection assay was carried out. As expected, intracellular ROS levels significantly increased in OA sf-MSCs donors compared with healthy ones (Figure 4), in agreement with a senescent phenotype in OA sf-MSCs.

### 2.5. High Expression of the Senescent Markers Cyclin-Dependent Kinase Inhibitors p21^WAF1/Cip1^ and p16^INK4A^ Is Detected in OA sf-MSCs

One common feature of a senescent state is a cell cycle arrest, characterized by the accumulation of cyclin-dependent kinase inhibitors (CDKIs) p21^WAF1/Cip1^ and p16^INK4A^. To verify their expression, western blot analysis was carried out on the total protein extract of healthy and OA sf-MSCs. As expected, the expression of p21^WAF1/Cip1^ (Figure 5a) and p16^INK4^ (Figure 5c) proteins was higher in OA sf-MSCs donors compared with healthy ones. Densitometric analysis of protein bands confirmed an upregulation of p21^WAF1/Cip1^ up to 30 folds in almost all the OA donors tested (Figure 5b) compared with healthy donors (*p* < 0.001) and an upregulation of p16^INK4^ protein up to 14 folds in OA sf-MSCs related to healthy donors (*p* < 0.001) (Figure 5d).

### 2.6. OA sf-MSCs Demonstrated a Variable Expression of COL2A1 mRNA

In order to evaluate the ability of OA sf-MSCs to differentiate toward a chondrogenic phenotype, MSCs from healthy and pathological donors were stimulated in vitro to chondrogenic differentiation for up to 21 days. H sf-MSCs showed a higher level of collagen type II (COL2A1) mRNA expression compared with the reference sample consisting of the first healthy donor (1H). On the contrary, OA sf-MSCs showed a very similar expression of COL2AI mRNA compared with H sf-MSCs, except for one donor (donor n°4) in which a 15-fold increase in the expression of mRNA COL2A1 was observed (*p* < 0.05) (Figure 6a).

SRY-box transcription factor 9 (SOX9) mRNA expression did not show any statistically significant difference between healthy and OA groups (Figure 6b), while the expression of aggrecan (ACAN) mRNA was slightly decreased in OA sf-MSCs compared with healthy cells (Figure 6c), suggesting a reduced ability of OA sf-MSCs in the synthesis of glycosaminoglycans (GAG), one of the main components of cartilaginous tissue.

### 2.7. OA sf-MSCs Showed a Reduced GAG Synthesis

To support a reduced synthesis of GAG, a specific GAG quantification assay was carried out in H and OA sf-MSCs at the end of in vitro chondrogenic differentiation. Figure 7 shows a statistically significant reduction of 0.5-fold in GAG in OA sf-MSCs compared with H sf-MSCs (*p* < 0.05), confirming a lower capacity of OA-MSCs in synthesizing GAG.

## 3. Discussion

OA is a joint disorder characterized by cartilage degradation, joint inflammation, subchondral bone remodeling, and fibrosis [10]. Current therapies for OA are mainly focused on treating symptoms of pain rather than counteracting the progression of the disease [10]. Recently, OA has been associated with the accumulation of senescent cells in joint tissues [12,19,22,23], but how senescence can affect each resident joint cell and its link with the onset and/or progression of OA is still poorly described.

With the progress of tissue engineering and regenerative medicine, MSCs have represented a promising candidate for cartilage repair and regeneration [24]. Widely studied for their immunomodulatory proprieties and their abilities to self-renew and to differentiate towards several lineages, including chondrocytes, no successful application based on the MSCs approach and the derived secretome has been developed for cartilage regeneration. However, most of the current research has been focused on the application of MSCs derived from bone marrow [25,26], adipose [27], and fetal-derived tissues [28,29], and just a few studies have investigated the potential therapeutic proprieties of sf-MSCs. Although sf-MSCs have demonstrated higher chondrogenic capabilities than other types of MSCs [30], their accumulation as senescent cells in synovial fluid and their correlation with OA progression have never been investigated. We have previously demonstrated a correlation of senescent sf-MSCs with age in healthy donors [3] in which the ability to differentiate toward a chondrogenic phenotype was mainly impaired in the oldest donors where the number of senescent cells was higher. In this study, we aimed to investigate the presence of senescent sf-MSCs in synovial fluid isolated from OA joints of adult young donors and their ability to differentiate toward a chondrogenic phenotype, assuming a compromised chondrogenic differentiation. A senescent condition in OA sf-MSCs in young donors could be responsible for premature aging which could support the onset and/or development of OA. Thus, the aim of the study was to verify the presence of senescent sf-MSCs isolated from OA joints and compare their chondrogenic capabilities with sf-MSCs isolated from healthy joints. Due to the high similarity between human and horse joints, the study was carried out on sf-MSCs isolated from OA and healthy equine joints. Furthermore, due to the heavy and repetitive mechanical load from high-speed training and racing [31], racehorses develop OA at a young age with symptoms clinically relevant and similar to human OA. The effect of non-fatal injuries in the racehorse industry is typically described in terms of “wastage”, which is the loss of horses due to injury or illness [32]. OA, and consequential lameness, is the most significant factor in the wastage of Thoroughbred racehorses [33].

We first characterized the isolated MSCs from healthy and OA joints for their mesenchymal and self-renewal phenotype, supposing a potential change of expression in OA sf-MSCs connected to the disorder. Results showed a positive expression for the mesenchymal markers CD90 and CD105 and for the self-renewal markers NANOG, SOX2, and OCT4, demonstrating their stemness and potential articular supportive proprieties even in pathological conditions. The presence of MSCs in each joint compartment was widely demonstrated, including synovial fluid, in healthy and pathological conditions [34,35]. Although the number of sf-MSCs was deeply increased in OA conditions [15], presumably to create an anti-inflammatory environment and to enhance cartilage healing, evidence of their efficacy was lacking, and their function remains unclear [36].

A strong correlation between the onset and/or development of OA and the accumulation of senescent cells in many joint compartments has been recently suggested [12,19,22,23]. Cellular senescence is characterized by a stable exit of the cell cycle and loss of replicative capacity, even in the presence of mitogenic stimuli [7,37]. Furthermore, the ability of these cells to secrete SASP components [9], which enhance the surrounding inflammatory microenvironment helpful for the OA progression [38], indicates that sf-MSCs are not quiescent cells but metabolically active [19]. Our findings regarding the proliferation activity of OA sf-MSCs and ROS production, agree with a state of cellular senescence. Indeed, the proliferation and doubling time assays showed a slower cell cycle in OA sf-MSCs compared with H ones. In detail, the flow cytometric analysis demonstrated a significant increase of cells in the G2/M phase in pathological cells compared with healthy ones, a condition compatible with cells damaged by oxidative and/or genotoxic stresses which trigger a DNA damage response (DDR) with a consequent delay in the cell cycle [39].

Senescence-associated β-galactosidase (SA-β-gal) is one of the most common assays used to demonstrate cellular senescence. It is based on the upregulation of the enzyme β-galactosidase in lysosomes, which has been demonstrated to increase in a senescent condition [40]. The reason why is still debated, but one accredited function is the link with the high secretion of the senescent cells, which need energy and raw material to synthesize and release the components of SASP [7,8]. Lysosomes and autophagy favored this digesting process and recycling of cellular material [41]. We performed TEM analysis to evaluate changes in the ultrastructure of OA and H sf-MSCs, with a particular interest in lysosomes. Unexpectedly, healthy sf-MSCs showed a cytoplasm rich in primary and secondary lysosomes and end products of the cellular digestion compared with OA sf-MSCs, in which the number of lysosomes was deeply reduced. Apparently, in contradiction with the SA-β-gal activity, our findings agree with reduced autophagy in senescent cells which is considered one of the hallmarks of senescence and aging [42,43,44]. Furthermore, a key role of autophagy in survival has been demonstrated, stem cell maintenance and tissue homeostasis, which sustains the stemness in stem cells and delays stem cell senescence [41,45]. These data could explain the high level of lysosomes in healthy sf-MSCs, which could be involved in autophagy as a fundamental mechanism to keep their self-renewal ability. On the contrary, the low activity of lysosomes in OA sf-MSCs could be involved in the progression of a senescent state.

Mitochondrial dysfunction is responsible for the release of ROS components, which can interact with protein and DNA, inducing cellular stress [46]. Elevated oxidative stress is another hallmark of cellular senescence and aging, and it has been linked to several age-related diseases, such as cardiovascular, pulmonary, and neurodegenerative disorders [47,48]. Our findings showed a significant increase of ROS in OA sf-MSCs compared with H sf-MSCs, confirming a senescent phenotype in OA cells which could amplify the progression of OA. Although not demonstrated in sf-MSCs, our data agree with the finding of elevated levels of ROS in aged and OA cartilage, both in human and animal models [49]. Furthermore, cells from human cartilage exposed to hydrogen peroxide showed hallmarks of chondrocyte senescence, such as reduced cell proliferation and lower synthesis of GAG [50,51].

ROS can induce DNA damage which triggers a DDR, followed by the activation of the p53/p21^WAF1/Cip1^ pathway and cell cycle arrest [7,9,37]. To confirm a senescent phenotype and a block in cell proliferation in OA sf-MSCs, the expression of the CDKIs p21^WAF1/Cip1^ and p16^INK4A^ was investigated in OA and H sf-MSCs. As expected, overexpression of both senescent markers was observed in OA sf-MSCs compared with H cells, in agreement with the previous data regarding cell proliferation and ROS detection and confirming a cell cycle arrest. Although the link between DDR, the activation of the p53/p21^WAF1/Cip1^ pathway, and cellular senescence is well documented [6,7,9,37], the function of p16^INK4A^ in cellular senescence is still unclear. Indeed, our data showed a significant increase of CDKI p16^INK4^ of almost 12 folds in one healthy sample. Chondrocytes and MSCs isolated from articular cartilage of OA patients showed an upregulation of p16^INK4A^ [18,34,52]. However, the somatic deletion of p16^INK4A^ in OA transgenic mice partially protects from cartilage degeneration [18]. Diekman and colleagues demonstrated that the inactivation of p16^INK4A^ in joint chondrocytes does not protect against OA [53]. Furthermore, the authors clearly demonstrated a strong correlation between p16^INK4A^ overexpression and the chronological age of mouse and human primary healthy chondrocytes [53].

Once the presence of cellular senescence has been demonstrated in OA sf-MSCs, our next step was to assess any correlation between the chondrogenic capacities of MSCs and the senescent state. Therefore, the expression of the chondrogenic markers COL2A1, SOX9, and ACAN, was analyzed in H and OA sf-MSCs stimulated to chondrogenic differentiation for up to 21 days. As expected, the expression of the chondrogenic transcription factor SOX9 did not show any significant difference between the two experimental groups. SOX9 is required to secure the commitment of skeletogenic progenitor cells to the chondrocyte lineage and to maintain this commitment throughout chondrocyte differentiation [3,28,54]. Supposing a low synthesis of COL2A1 mRNA in OA samples, we were surprised to observe a very similar expression of COL2A1 mRNA compared with H sf-MSCs except in one donor, in which a 15-fold upregulation of COL2A1 mRNA was observed. These data indicate a high variability of the chondrogenic marker COL2A1 in OA sf-MSCs, which does not allow us to obtain any reasonable conclusion on the synthesis of collagen type II marker. We think that the variability observed is likely due to a low number of donors involved in the study. ACAN mRNA expression showed downregulation in OA sf-MSCs compared with H cells, supported by DMMB assay, specific for GAG quantification, which enforces the reduced chondrogenic potential of OA sf-MSCs. Supposing to observe a reduced synthesis of the chondrogenic markers in OA sf-MSCs, we speculate that the unexpected expression of COL2A1 marker, such as H sf-MSCs, could be related to a fibrotic condition which typically occurs in the articular joint during OA progression and plays a critical role in OA pathogenesis as well as cartilage destruction [55,56]. The reduced synthesis of GAG supports this hypothesis, but further investigations, based on a higher number of donors, are needed to confirm the assumption.

## 4. Materials and Methods

### 4.1. Isolation of Equine sf-MSCs

The ethical committee of the University of Teramo (3 October 2018) and the Italian Ministry of Health (approval No. 0018640-P-2/0772018) approved sample collection procedures. After obtaining the informed consent from the owners, synovial fluid (sf) was aseptically collected from tibiotarsal joints of horses ranging from 3 to 14 years old and stored in BD Vacutainer^®^ tubes with EDTA. The age, breed, sex, and number of samples are summarized in Table 2. The diagnosis of OA was based on clinical and radiological evaluation of the subjects included in the study. A minimum of 5 mL of clear and blood-free sf was obtained from each tibiotarsal joint. Sf was diluted with phosphate-buffered saline (PBS) for up to five volumes, and it was subsequently filtered through a 70 μm nylon filter (Cell Strainer, BD Falcon, Leipzig, Germany) to remove debris and centrifuged at 160× *g* for 5 min at room temperature (RT). Pellets were resuspended in Minimal Essential Medium (MEM, Life Technologies, Monza, Italy), supplemented with 10% fetal bovine serum (FBS, Life Technologies, Monza, Italy), and plated in a 25 cm^2^ culture flask, at 37 °C in 5% CO_2_ for 7 days. Cells adhering to the bottom of the flask were subsequently detached and cultured in the same conditions. Cells from passage 1 to passage 6 were utilized for all the experiments of the study. All the experiments were carried out using cells of healthy and pathological donors at the same passage to minimize any difference due to the different cell division steps. Table 2 summarizes the number of donors (*n* = 8), age, and gender included in the study. Cells obtained were divided into two groups: cells isolated from healthy joints (H) and cells isolated from OA joints (OA).

### 4.2. mRNA Expression of Mesenchymal and Self-Renewal Markers by RT-PCR

Total RNA was obtained with RNeasy Mini Kit (Invitrogen, Thermo Fisher Scientific, Waltham, MA, USA) starting from cellular pellets. 100 ng of total RNA was reverse transcribed into first-strand cDNA using SuperScriptTM III One-Step RT-PCR System (Invitrogen, Thermo Fisher Scientific, Waltham, MA, USA). The PCR reaction was performed by Phusion Green Hot Start II High-Fidelity DNA Polymerase kit (Thermo Fisher Scientific, Waltham, MA, USA). The equine glyceraldehyde-3-phosphate dehydrogenase gene (GAPDH) was used as a housekeeping gene for amplification control during the PCR assay. The primer sequences were designed by Primer design Software (Primer3-based OligoPerfect, Thermo Fisher Scientific, Waltham, MA, USA). The amplified DNA was then electrophoresed on a 2% agarose gel and visualized by ethidium bromide staining; images were acquired by Image Station 2000R (Kodak, New York, NY, USA). The primer sequences used are shown in Table 3.

### 4.3. BrdU Proliferation Assay

Cell proliferation was assayed by BrdU assay kit (Roche, Basel, Switzerland) according to the manufacturer’s instructions. Briefly, sf-MSCs, isolated from healthy and OA synovial fluid, were seeded into a 96-well culture plate with MEM containing 10% FBS at the density of 8 × 10^3^ cells/well. After 24 h, the medium was changed with a fresh one containing 10 μM BrdU for 48 h at 37 °C. After the removal of the labeling solution, cells were fixed and incubated with anti-BrdU conjugated with peroxidase antibody for 90 min at RT. After three washing steps in PBS, a tetramethyl-benzidine (TMB) substrate solution was added for 10 min at RT, and the reaction was stopped with 1 M H_2_SO_4_. The optical density was measured using a spectrophotometer microplate reader (LT-4000 Microplate reader, Labtech Ltd., Heathfield, UK) at a wavelength of 450 nm and a reference wavelength of 690 nm. Data were expressed in optical density (O.D.) and were represented as mean values ± SD. Results of each donor were shown. The experiments were repeated at least four times, and each experimental point was done in triplicate.

### 4.4. Doubling Time Assay

Sf-MSCs isolated from all donors were seeded at the density of 5 × 10^3^ cells/cm^2^ in 25 cm^2^ flasks and allowed to grow until reaching 90% of confluence. At this point, cells were detached with enzyme digestion, counted by automated cell countess (CountessTM Automated Cell Counter, Invitrogen, Thermo Fisher Scientific, Waltham, MA, USA), and reseeded at the same density as previously described. The procedure was repeated from passage p1 to p5. Cell-doubling time (DT) was calculated from counts for each passage according to the following two formulae [57]:CD = ln(Nf/Ni)/ln(2)(1)
DT = CT/CD(2)
where Nf and Ni are the final and initial number of cells, respectively, and CT is the cell incubation time expressed in days.

### 4.5. Cell Cycle Analysis

Flow cytometric analysis of the cell cycle was performed using propidium iodide (PI)/RNase A staining according to standard procedures, as described previously [58]. Briefly, the Sf-MSCs were detached from the surface of the flask with enzyme digestion for 3 min at room temperature and then collected and centrifuged at 300× *g* for 5 min. Cell pellets were resuspended in PBS and counted by a hemocytometer. A total of 2.5 × 10^5^ cells was centrifuged and then resuspended in 70% ethanol overnight at 20 °C. Cells were then centrifuged at 300× *g* for 5 min, and the pellets obtained were washed twice in PBS and resuspended in a solution of propidium iodide for at least 30 min. Samples were analyzed on an FC500 flow cytometer (Beckman Coulter, Indianapolis, IN, USA) with the appropriate software (version 2.2, CXP, Beckman Coulter). At least 15,000 events per sample were acquired.

### 4.6. Reactive Oxygen Species (ROS) Detection Assay

The 6-carboxy-2′,7′-dichlorodihydrofluorescein diacetate (carboxy-H2DCFDA) fluorescent labeling was adopted to quantify the total intracellular production of reactive oxygen species (ROS). Briefly, 10^4^ cells/well were seeded into a 96-well culture plate in MEM containing 10% FBS. After 24 h, the medium on adherent cells was replaced with a fresh medium supplemented with a 5 μM carboxy-H2 DCFDA ROS detection probe (Thermo Fisher Scientific, Monza, Italy). The incubation was performed for a further 3 h at 37 °C. The fluorescent signal was measured using a fluorimeter microplate reader (Glomax, Promega Corporation, Madison, WI, USA) at a fluorescence excitation of 492 nm and at fluorescence emission of 517 nm.

### 4.7. Western Blot Analysis of Senescent Markers

Total protein lysate in OA-MSCs and H-MSCs was extracted using RIPA-modified cell lysis buffer (Pierce, Thermo Fisher Scientific, Monza, Italy) supplemented with 25 μmol/L protease inhibitor cocktail (Pierce, Thermo Fisher Scientific, Monza, Italy) and 1 μL of β-mercapto-ethanol (Sigma-Aldrich, St. Louis, MS, USA). The amount of protein obtained from each sample was quantified by Bradford assay (Sigma-Aldrich, St. Louis, MS, USA), and 15 μg of lysate was loaded and separated with 4–12% sodium dodecyl sulfate-polyacrylamide gel electrophoresis (SDS-PAGE), followed by transfer onto a nitrocellulose membrane (GE Healthcare, Amersham, UK). Subsequently, the membranes were incubated with 5% BSA (blocking reagent) to remove the nonspecific binding proteins, followed by incubation with the primary antibodies against rabbit anti-p21^WAF1/Cip1^ antibody (Cell Signaling Technologies, Euroclone, Milan, Italy), rabbit anti-p16^INK4A^ antibody (Cell Signaling Technologies, Euroclone, Milan, Italy) and anti-actin antibody (Millipore Merck, Darmstadt, Germany), diluted 1:1000 in blocking reagent at 4 °C, overnight. After washes with TBS-tween buffer, samples were incubated with HRP-linked anti-rabbit or anti-mouse IgG secondary antibodies, diluted 1:2000 in TBS-tween buffer (Sigma Aldrich, St Louis, MS, USA). The antibody signal was visualized by the enhanced chemiluminescence system (Pierce, Thermo Fisher Scientific, Monza, Italy). Images were obtained by using IBright Western Blot Imaging System (Thermo Fisher Scientific, Waltham, MA, USA). Finally, the densitometric analysis was performed using Image J-2 software version 2 (National Institutes of Health, Bethesda, MD, USA). The intensities of the specific protein bands were corrected for equal actin loading, and they were expressed as a relative value compared with the intensity of the first healthy donor (1H). Data showed the median ± SD of three independent experiments.

### 4.8. Transmission Electron Microscopy (TEM)

Sf-MSCs cultured in monolayers were fixed with 2.5% (*v*/*v*) glutaraldehyde in 0.1 M cacodylate buffer for 2 h and post-fixed with a solution of 1% (*w*/*v*) osmium tetroxide in 0.1 M cacodylate buffer. The cells were then embedded in epoxy resin after a graded-acetone serial dehydration step. The embedded cells were sectioned into ultrathin slices, stained by uranyl acetate solution and lead citrate, and then observed by transmission electron microscope CM10 Philips (FEI Company, Eindhoven, The Netherlands) at an accelerating voltage of 80 kV. Images were recorded by Megaview III digital camera (FEI Company, Eindhoven, The Netherlands). TEM analysis was performed in duplicate for cells obtained from each healthy and pathological sample.

### 4.9. Chondrogenic Differentiation

Sf-MSCs were cultured in MEM, supplemented with 10% fetal bovine serum (FBS) (Sigma-Aldrich, St. Louis, MO, USA), and then they were grown into 3D micromass culture. Briefly, 5 × 10^5^ cells were pelleted (10 min, 160× *g*) per tube (Sarstedt, Neumbrecht, Germany) and kept in 1 mL chondrogenic medium consisting of MEM supplemented with 2% FBS, 100 nM dexamethasone (Sigma-Aldrich, St. Louis, MS, USA), 100 μg/mL ascorbate-2-phosphate (Sigma-Aldrich, St. Louis, MS, USA), ITS (6.25 μg/mL insulin, 6.25 μg/mL transferrin, 6.25 μg/mL selenium) (Gibco, Thermo Fisher Scientific, Monza, Italy), and 10 ng/mL of TGFβ3 (Millipore, Milan, Italy) for 21 days at 37 °C and 5% CO_2_. At the end of each treatment, cells were collected, and RNA was extracted for quantitative real-time PCR (qRT-PCR) to evaluate the expression of SOX9, COL2A1, and ACAN mRNA chondrogenic markers.

### 4.10. Real-Time PCR

The chondrogenic mRNA marker expression was analyzed by Real-Time PCR (7500 Applied Biosystems, Life Technologies, Monza, Italy). For mRNA extraction and cDNA amplification, the same protocol described in paragraph 4.2 was used. For mRNA quantification, the Powerup SYBR master mix kit (Life Technologies, Thermo Fischer Scientific, Monza, Italy) was used in combination with the specific primers for the chondrogenic markers shown in Table 4. Relative gene expression levels were normalized to those of equine glyceraldehyde 3-phosphate dehydrogenase. Data are presented as fold changes relative to levels of the reference sample by using formula 2^−ΔΔCT^, as recommended by the manufacturer (User Bulletin number 2 P/N 4303859; Applied Biosystems,, Foster city, CA, USA). The quantitative analysis of mRNA expression was performed in triplicate for each donor. The results were expressed as fold changes compared with the first healthy donor (1H) chosen as a reference control. Only the samples stimulated to chondrogenic differentiation were investigated for the expression of chondrogenic markers.

### 4.11. GAG Quantification

At the end of in vitro chondrogenic differentiation, performed as described in paragraph 4.8, 3D masses were analyzed for extracellular chondrogenic GAG deposition. The assay was performed by measuring the reaction between GAGs and 1,9-dimethylmethylene blue (DMMB) reagent (Sigma–Aldrich). Samples were digested overnight with 0.3 mg/mL papain solution in a phosphate/EDTA buffer, pH 6.5, at 65 °C. At the end of incubation, DMMB solution was added to the matrix extraction to develop a GAG-dye complex and spectrophotometrically read at 525 nm (Glomax, Promega Corporation, Madison, WI, USA). Total GAG content was extrapolated referring to a standard curve set up with shark chondroitin sulfate (Sigma-Aldrich, MS, USA). The experiment was repeated three times, and each sample was in triplicate. The final values were normalized to the number of cells used for each sample and represented as relative values ± SD compared with the first healthy donor (1H) chosen as reference control.

### 4.12. Statistical Analysis

All statistics were performed using Prism 6 (GraphPad, San Diego, CA, USA). Student’s *t*-test was used to compare the difference between the two groups, OA sf-MSCs vs. H sf-MSCs. The differences were considered significant at *p* < 0.05.

## 5. Conclusions

The onset and progression of OA is a complex process. Our data demonstrated the presence of senescent MSCs in the synovial fluid of the OA joint, which could have a great influence on the progression of OA. The reduced chondrogenic abilities and the production of SASP components could strongly transform the surrounding microenvironment and support OA development. This condition could reduce the efficacy of therapies based on MSCs and their secretome.

Therapeutical approaches targeting senescent cells are emerging as a promising strategy to delay OA progression. Indeed, senotherapy, based on senolytic drugs which induce apoptosis in senescent cells and senomorphic drugs which disturb the SASP release, is a novel and attractive candidate for targeting OA and slowing its progression.

## Figures and Tables

**Figure 1 ijms-24-03109-f001:**
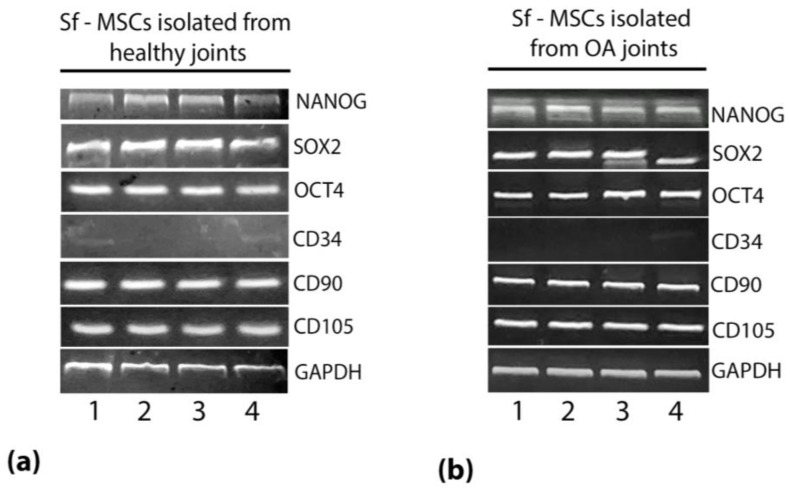
Representative images of agarose gel electrophoresis of RT-PCR products showing the positive expression of the self-renewal markers NANOG, SOX2, and OCT4 and of the mesenchymal surface marker CD90, CD105 and the lack of expression of the hematopoietic surface marker CD34, in healthy (**a**) and OA (**b**) sf—MSCs. Cells from different donors were tested (*n* = 4).

**Figure 2 ijms-24-03109-f002:**
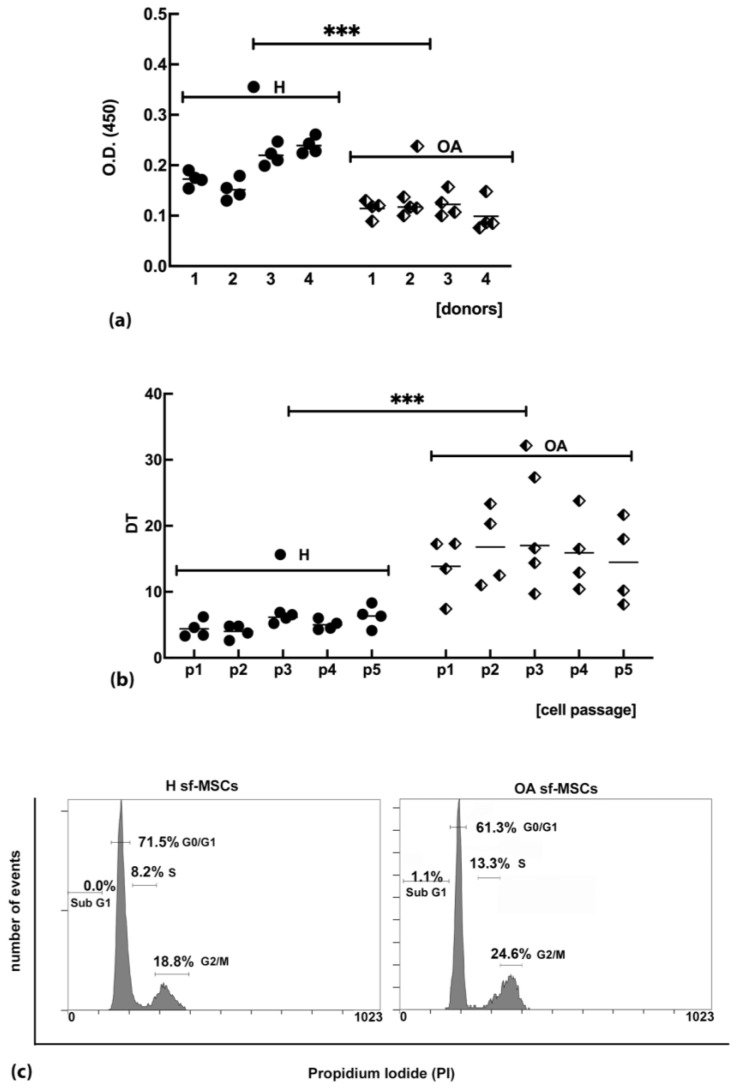
(**a**) BrdU cell proliferation assay of sf-MSCs, isolated from healthy (H) and osteoarthritic (OA) joints. Cell proliferation is significantly lower in OA compared with H sf-MSCs. The mean value (black line) corresponding to each group (*n* = 4) is represented. Data are expressed in optical density (O. D. 450 nm). (**b**) doubling time assay of sf-MSCs, isolated from H and OA joints. Doubling time is expressed in days, and it is significantly higher in OA compared with H sf-MSCs, in all the donors tested. The mean value (black line) corresponding to each donor (*n* = 4) is represented. Data are grouped under H and OA groups. (**c**) Representative results of the cell cycle distribution of sf-MSCs isolated from H and OA joints and analyzed by flow cytometry based on propidium iodide staining. The percentage of cells in each phase of the cell cycle is shown in a bar graph form—number of events: cell frequency; Propidium iodide: DNA staining fluorochrome. Student’s *t*-test was used to calculate statistical significance between H and OA groups; *** *p* < 0.001.

**Figure 3 ijms-24-03109-f003:**
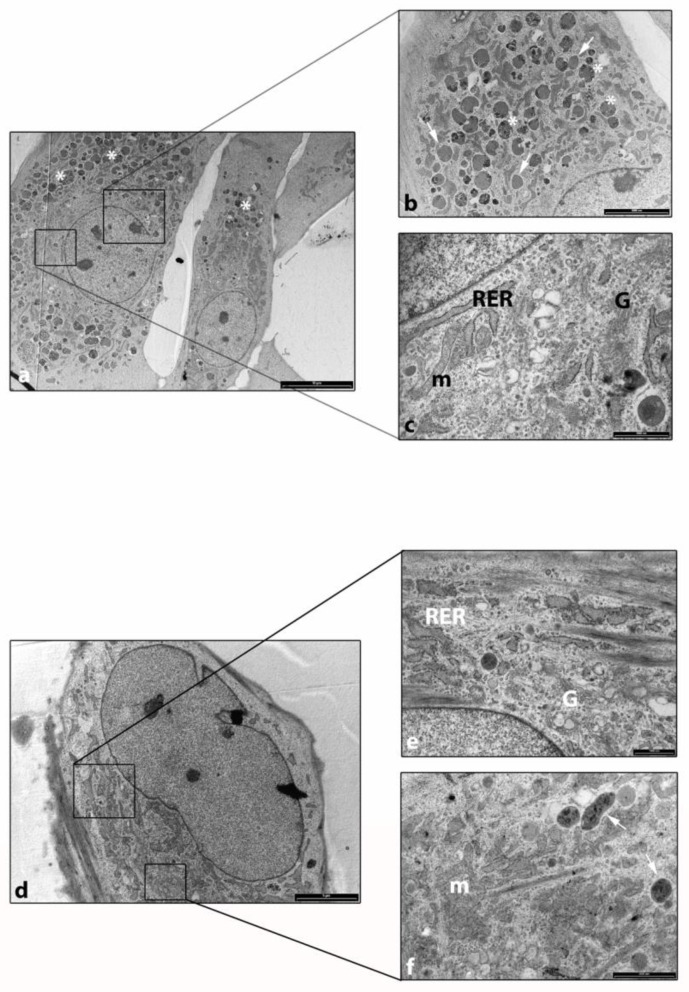
(**a**) Representative images of TEM analysis of H sf-MSCs. The cytoplasm is characterized by the presence of several primary and secondary lysosomes (white *) as well as their intermediate and final products of cellular digestion (bar: 10 μm). (**b**) Cytoplasmic detail of H sf-MSCs showing primary (white arrow), secondary (white *), and end products of cellular digestion (bar: 5000 nm); (**c**) Detail of cytoplasmic area in which rough endoplasmic reticulum (RER), Golgi complex (G) and mitochondria (m) are observed (bar: 1000 nm). (**d**) Representative images of TEM analysis of OA sf-MSCs. The cytoplasm is almost free of primary and secondary lysosomes (bar: 5 μm); (**e**) Detail of cytoplasm in OA sf-MSCs showing RER and several vesicles connected to the activity of Golgi complex (G) (bar: 2000 nm); (**f**) Detail of cytoplasm in OA sf-MSCs showing a few primary lysosomes (white arrow) and mitochondria (m) (bar: 1000 nm).

**Figure 4 ijms-24-03109-f004:**
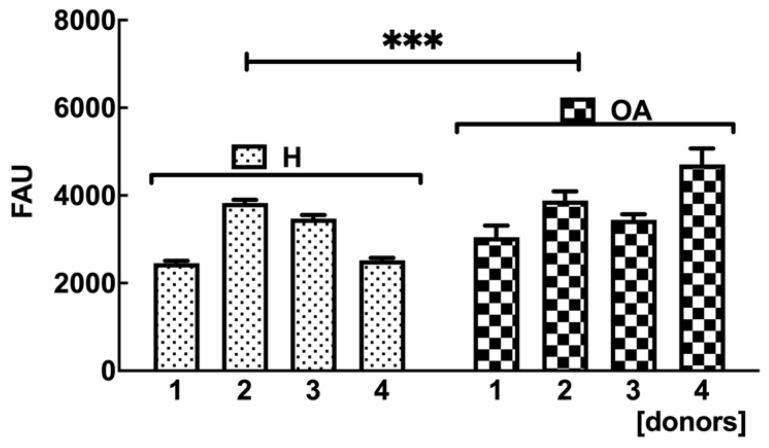
Total intracellular production of ROS detected by carboxy-H2DCFDA fluorescent assay in H and OA MSCs (for each group *n* = 4). Results are presented in FAU (fluorescent arbitrary units). The value of each donor is represented as mean ± SD. Student’s *t*-test was used to calculate statistical significance between H and OA groups; *** *p* < 0.001.

**Figure 5 ijms-24-03109-f005:**
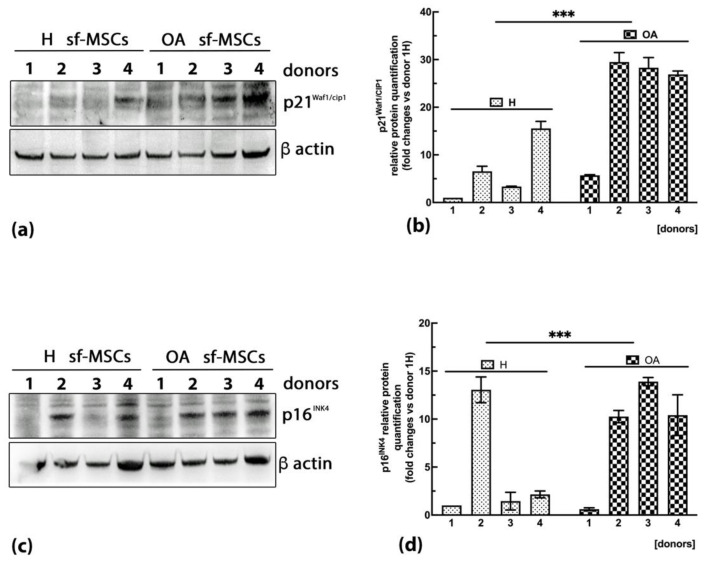
(**a**) representative western blot images showing p21^WAF1/Cip1^ expression in H and OA sf-MSCs. Results of all donors of each experimental group are shown (*n* = 4). (**b**) Relative amounts of p21^WAF1/Cip1^ expression normalized to the intensity of β-actin and represented as fold increase relative to first donor of healthy MSCs (1H). Western blot was performed in duplicate, and the relative quantification is expressed as mean value ± SD. (**c**) representative western blot images showing p16^INK4^ expression in H and OA sf-MSCs. Results of all donors of each experimental group are shown (*n* = 4). (**d**) Relative amounts of p16^INK4^ expression normalized to the intensity of β-actin and represented as fold increase relative to the first donor of healthy MSCs (1H). Western blot was performed in duplicate, and the relative quantification is expressed as mean value ± SD. *** represents a significant difference compared with H sf-MSCs, *p* < 0.001.

**Figure 6 ijms-24-03109-f006:**
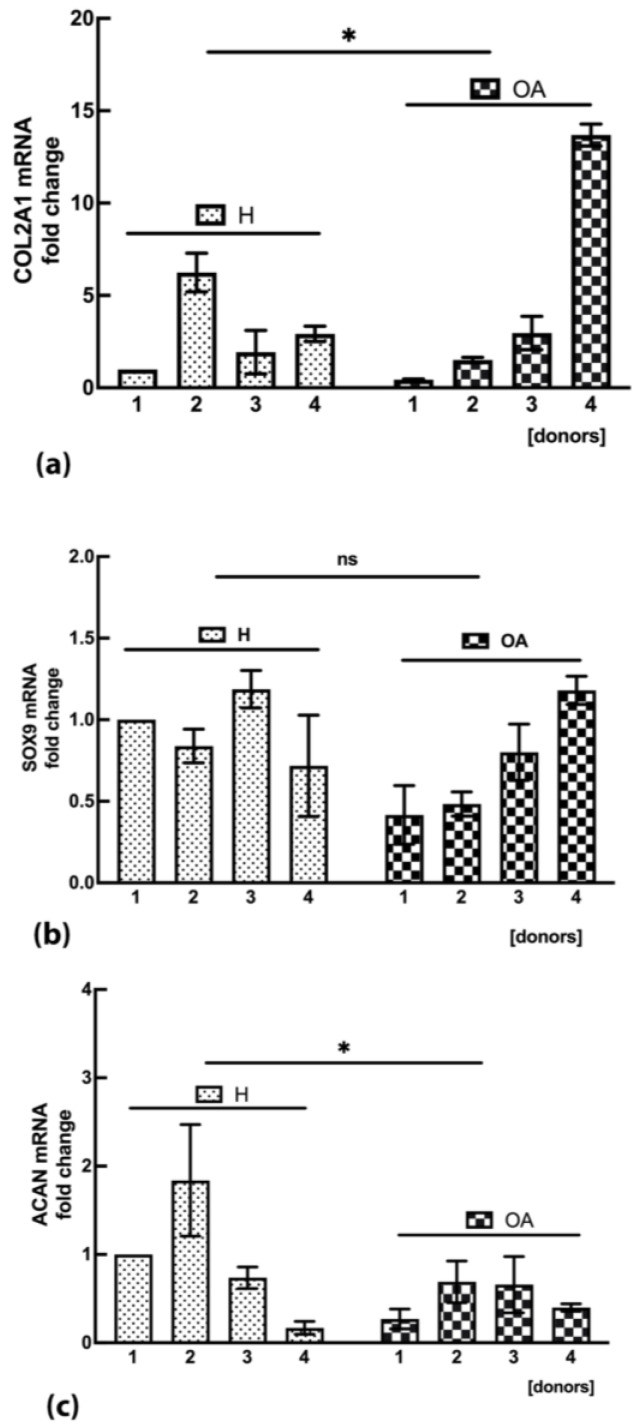
(**a**) COL2A1 mRNA expression in H and OA sf-MSCs, induced to in vitro chondrogenic differentiation up to 21 days and measured by Real-Time PCR. Results are expressed as fold change compared with the first donor of healthy group (1H). Real-time was performed in triplicate, and the results for each donor are represented as the mean value ± SD. (**b**) SOX9 mRNA expression in H and OA sf-MSCs, induced to in vitro chondrogenic differentiation up to 21 days and measured by Real-Time PCR. Results are expressed as fold change compared with the first donor of healthy group (1H). Real-time PCR was performed in triplicate, and the results for each donor are represented as the mean value ± SD. (**c**) ACAN mRNA expression in H and OA sf-MSCs, induced to in vitro chondrogenic differentiation up to 21 days and measured by Real-Time PCR. Results are expressed as fold change compared with the first donor of healthy group (1H). Real-time PCR was performed in triplicate, and the results of each donor are represented as the mean value ± SD. (ns) represents the lack of significant difference compared with H group. * represents a significant difference compared with H group, *p* < 0.05.

**Figure 7 ijms-24-03109-f007:**
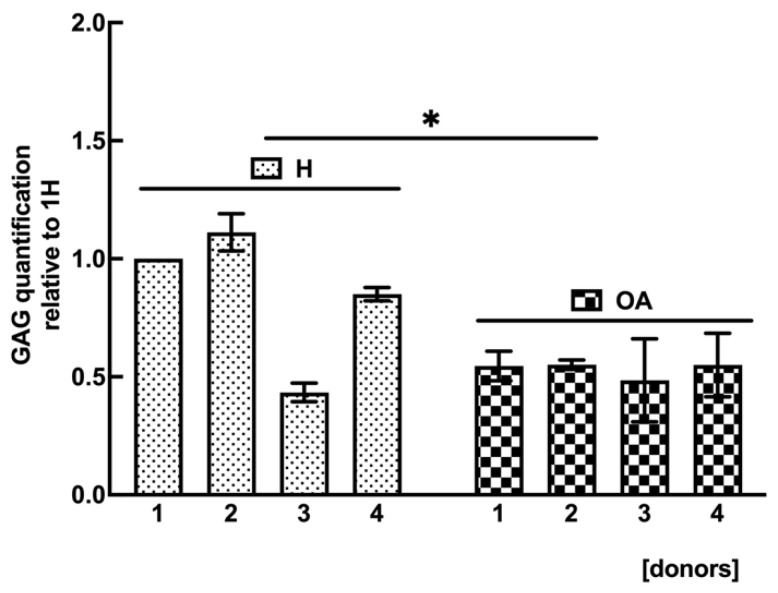
Relative GAG quantification obtained from micromasses of H and OA sf-MSCs in vitro stimulated to chondrogenic differentiation up to 21 days. Extracts were quantified by DMMB assay, and the data obtained were normalized to the number of cells utilized for each donor and represented as relative value compared with the first healthy donor (1H). DMMB assay was performed in triplicate, and the results of each donor are represented as the mean value ± SD. * represents a significant difference compared with H group, *p* < 0.05.

**Table 1 ijms-24-03109-t001:** Flow cytometry data of cell cycle distribution in healthy (H group) and osteoarthritic sf-MSCs (OA group). Values corresponding to each donor are reported and organized in the experimental group. The experiment was repeated in triplicate for each donor, and the results are expressed as percentage ± SD. * represents a significant difference from 1H donor; *p* < 0.05.

**H Group**	**G0/G1(%)**	**S Phase (%)**	**G2/M (%)**
1H	82.5 ± 6.4	4.4 ± 2.4	12.8 ± 5.7
2H	65.5 ± 1.7 *	3.7 ± 1.4	15.5 ± 1.3
3H	75.5 ± 5.7	7.8 ± 0.4 *	15.2 ± 5.0
4H	59.7 ± 6.4 *	13.9 ± 6.3 *	24.3 ± 9.3 *
**OA Group**	**G0/G1(%)**	**S Phase (%)**	**G2/M (%)**
1 OA	63.7 ± 3.4 *	10.4 ± 4.0 *	25.1 ± 0.7 *
2 OA	62.6 ± 1.6 *	17,8 ± 3.4 *	17.1 ± 2.3 *
3 OA	52.9 ± 9.7 *	13.5 ± 3.4 *	32.6 ± 13.2 *
4 OA	56.1 ± 1.4 *	24.0 ± 2.9 *	16.9 ± 0.7 *

**Table 2 ijms-24-03109-t002:** Profiles of sf samples obtained from the Healthy (H) and Pathological (OA) joints of horses.

**Group H**	**Breed**	**Age (Years)**	**Gender**
1 H	Thoroughbred	3	M
2 H	Standardbred	6	F
3 H	Standardbred	10	F
4 H	Standardbred	14	F
**Group OA**	**Breed**	**Age (Years)**	**Gender**
1 OA	Thoroughbred	3	M
2 OA	Thoroughbred	6	F
3 OA	Standardbred	8	M
4 OA	Pleasure riding horse	10	F

**Table 3 ijms-24-03109-t003:** RT—PCR primer sequences.

Name	Forward	Reverse	bp
*CD90*	5′-ATGAGAATACCACCGCCACA-3′	5′-AGTTTGTCTCGGAGCACAGA-3′	262
*CD105*	5′-TCAGGTCCCCAACACTAACC-3′	5′-AGTCTTGTTCGTGCTGAGGA-3′	148
*CD34*	5′-CCTTGCCCAGTCTGAGGTTA-3′	5′-GTCTTGCGGGAATAGTGCTG-3′	172
*NANOG*	5′-TCTCTCCTCTGCCTTCCTCC -3′	5′-TCTGCTGGAGGCTGAGGTAT-3′	225
*OCT4*	5′-GGTACGAGTGTGGTTCTGCA-3′	5′-ACCGAGGAGTACAGCGTAGT-3′	192
*SOX2*	5′-GCCCTGCAGTACAACTCCAT-3′	5′-GACTTGACCACCGAACCCAT-3′	128
*GAPDH*	5′-TGCCCCAATGTTTGTGATGG-3′	5′-CACTGTGGTCATGAGTCCCT-3′	154

**Table 4 ijms-24-03109-t004:** Real-Time—PCR primer sequences.

*SOX9*	5′-GAACAGCCCGTCTACACACA-3′	5′-GCCACTGATTCGCAACAAGG-3′	235
*COL2A1*	5′-CTGGCAAGCAAGGAGACAGA-3′	5′-CCATTAGCGCCATCTTTGCC-3′	292
*ACAN*	5′-TCATCTAGAGCCCACTGCCT-3′	5′-AGTCCACCGAGGTCCTCTAC-3′	234
*GAPDH*	5′-TGCCCCAATGTTTGTGATGG-3′	5′-CACTGTGGTCATGAGTCCCT-3′	154

## Data Availability

Data are contained within the article. For any further detail fell free to contact the corresponding author.

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
