# Peer review of "Implication of Cellular Senescence in Osteoarthritis: A Study on Equine Synovial Fluid Mesenchymal Stromal Cells"

_ijms, 2023, doi:10.3390/ijms24043109_

Round 1
Reviewer 1 Report
The work seems to me well written and I can't find anything else to addAuthor Response
We are really grateful to the review for reading and appreciating the study.
Reviewer 2 Report
The authors of the manuscript aimed at determining the presence and accumulation of senescent sf-MScs in OA joints and their potential contribution in disease progression using an equine model in view of the fact that there is a high similarity in the anatomy and biomechanics between human and horse knee joint.
The manuscript reports interesting results as the deepening of knowledge on the animal MSCs could provide important information for translational research. However, the manuscript could be improved by making some changes and reporting some information.
Line 446: cells from passage 1 to passage 6 were used for all the experiments of the study: the results are reported per horse and not per passage, it is not clear whether the reported results are overall for all passages or refer to a single passage. It is important to specify that all the analyses carried out on the different horses were carried out on the same cellular passages as there could be differences between the various passages, for example, in terms of senescence and differentiation capacity.
Has any staining been carried out to check for chondrogenic differentiation?
Was the chondrogenic mRNA markers expression also evaluated in undifferentiated cells using them as a control? This should be specified.
Minor issues
Line 95: replace “horsed” with “horses”.
Line 97: replace “a equine” with “an equine”.
The acronyms NANOG and OCT4 should be reported uniformly (uppercase or lowercase) in the text and in Figure 1.
Line 194: insert period at the end of the sentence.
Line 210: replace “detail” with “Detail”.
Line 307: replace “relative” with “Relative”.
Line 441, 520, 523, 575: replace “g” with “xg”.
Line 519: the acronym sf-MSCs should be reported uniformly in the text.
Line 606: replace ml” with “mL”.
Author Response
Comments and Suggestions for Authors
The authors of the manuscript aimed at determining the presence and accumulation of senescent sf-MScs in OA joints and their potential contribution in disease progression using an equine model in view of the fact that there is a high similarity in the anatomy and biomechanics between human and horse knee joint.
The manuscript reports interesting results as the deepening of knowledge on the animal MSCs could provide important information for translational research. However, the manuscript could be improved by making some changes and reporting some information.
Q: Line 446: cells from passage 1 to passage 6 were used for all the experiments of the study: the results are reported per horse and not per passage, it is not clear whether the reported results are overall for all passages or refer to a single passage. It is important to specify that all the analyses carried out on the different horses were carried out on the same cellular passages as there could be differences between the various passages, for example, in terms of senescence and differentiation capacity.
A: Thank you for the question and suggestion to improve the study. The experiments were carried out trying to respect the same passage in each donor, to better compare the results. In our previous experience between healthy donors (Mazzotti et al., doi: 10.3390/cells8101116), we do not see any significant differences from passage 1 to passage 13 in replication rate, duplication time and cell cycle analysis. We agree that in the group with pathological donors there could be significant differences, this is the reason why we performed all the experiments trying to use a low number of passages as much as possible. We specified the use of the same cell passage in healthy and pathological donors in line 454-455, according to your suggestion.
Q: Has any staining been carried out to check for chondrogenic differentiation?
A:Thank you for the question. In vitro working with senescent cells is quite hard due to a reduced/lack of proliferation. To perform any staining procedures specific for chondrogenic differentiation (alcian blue and/or safranin staining procedures) we need a high number of cells to prepare micromasses and then to process samples for light microscopy and staining procedures. Due to reduce efficiency of the staining technique compared to Real Time PCR, we decided to first perform PCR that allow to obtain results even with a low number of cells. The staining procedure would have supported the Real Time PCR data, but to avoid using cells over passage 6 we decided to utilize cells for the other experiments to complete the study.
Q:Was the chondrogenic mRNA markers expression also evaluated in undifferentiated cells using them as a control? This should be specified.
A:Yes, the expression of chondrogenic markers was also evaluated between undifferentiated samples, obtaining data showing a low expression of chondrogenic markers, as supposed, and not statistically significant between the two experimental groups. Based on our previous experience (Mazzotti et al., doi: 10.3390/cells8101116) and taking into consideration the low/lack of proliferation of senescent cells, we gave priority to experiments based on differentiated samples, where was more interesting to detect any differences. We specified the samples utilized in line 605-606.
Minor issues
A:Thank you very much for all the suggestions and corrections you underlined in the minor issues. You will find any correction in the text with a red color.
Line 95: replace “horsed” with “horses”.
Line 97: replace “a equine” with “an equine”.
The acronyms NANOG and OCT4 should be reported uniformly (uppercase or lowercase) in the text and in Figure 1
Line 194: insert period at the end of the sentence.
Line 210: replace “detail” with “Detail”.
Line 307: replace “relative” with “Relative”.
Line 441, 520, 523, 575: replace “g” with “xg”.
Line 519: the acronym sf-MSCs should be reported uniformly in the text.
Line 606: replace ml” with “mL”.

Reviewer 3 Report
Only few things:
- I suggest highlighting more in the Introduction and in the Discussion the innovations demonstrated in this manuscript compared to the previous paper (Ref 3) from which I think the idea for the project started
- Standardize the characters in the references (e.g. ref 36)
- Reduce self-references
- It is stated that the research was done without external funds. How is it possible to have done such a large and in-depth study without funds?
Author Response
Comments and Suggestions for Authors
Only few things:
Q: I suggest highlighting more in the Introduction and in the Discussion the innovations demonstrated in this manuscript compared to the previous paper (Ref 3) from which I think the idea for the project started
A: Thank you very much for your important suggestion. We have changed and hopefully improved the introduction section according to your suggestion. In the text you will find any added and changed sentence written in red (Introduction lines 91-94; Discussion lines from 335 to 342).
Q:Standardize the characters in the references (e.g. ref 36)
A: we checked and standardized the references. Thank you.
Q: Reduce self-references
A: we have reduced the self-references in agreement with your suggestion (substituting the n° 27, 28 and 57 with more pertinent references), leaving the ones we think are appropriate for the topic of the study (n° 3 and 5).
Q: It is stated that the research was done without external funds. How is it possible to have done such a large and in-depth study without funds?
A: Although there is not a specific grant on this project, every year the University of Bologna supports the professors and researchers in their research activity with a small donation (ex-RFO). The authors from the University of Bologna decided to put together their efforts to support this study. In the text, in the section relative to funding we changed the sentence expressing gratitude to the University of Bologna.

Reviewer 4 Report
The article by Teti et al entitled "Implication of Cellular Senescence in Osteoarthritis. A Study on Equine Synovial Fluid Mesenchymal Stromal Cells" describes the differences at the level of senescence markers and differentiation between MSCs derived from synovial fluid of healthy horses and those with osteoarthritis. Although there are studies associating synovial MSC senescence with osteoarthritis, research on this topic is of interest because of its potential clinical implications for the treatment of this pathology.
Remarks
- It is surprising that since this is a study on senescence, no data on β-galactosidase activity have been presented. In my opinion, it would have been interesting to measure this activity in the two groups of MSCs studied to complete the results.
- The design of the study is not entirely adequate. In each group the authors have introduced animals of very different ages ranging from 3 to 14 years old. As they themselves have shown in previous studies (Mazzotti et al., 2019 doi:10.3390/cells8101116), age influences the differentiation and viability of sf-MSCs in horses. Therefore, to the variability existing between different individuals in this type of studies, we must add the variability originated by the age difference. This makes it difficult to interpret the results. This may explain, for example, that animal No. 1 of the OA group, which is the youngest, presents very low values of P21 and P16 with respect to the older animals of the same group (Fig. 5) and less ROS production, with values very similar to those of the H group (Fig. 4).
- The statistics in Figure 6a and 6c should be revised or explained as it has been done. In both figures, according to the values represented, there is a large dispersion of data within each figure, making it difficult to understand that the differences are significant. In the case of Figure 6a, in the OA group there is only one animal with very high values with respect to the others. In the rest, it can be seen that the COL2A1 expression of No. 1 is the lowest of all, including those of group H, and that of animals 2 and 3 are at the average or lower than those of group H. Therefore, with these data it is very speculative to say that COL2A1 expression is higher in OA with respect to H. In my opinion, the study of chondrocyte marker gene expression shows no clear differences between the two groups of animals and should not be highlighted in the manuscript. Only with a larger number of donors of more homogeneous age, they could have more reliable data.
- Minor remarks:
On line 281 it would be convenient to put "glycosaminoglycans (GAG)", due to the fact that GAG has not been defined before.
In the figure legends it is better to put the meaning of the statistical differences at the end of the legend and not to repeat the same text several times. For example, in the legend of Figure 5, line 260 repeats the same text as lines 265-266.
Author Response
Comments and Suggestions for Authors
The article by Teti et al entitled "Implication of Cellular Senescence in Osteoarthritis. A Study on Equine Synovial Fluid Mesenchymal Stromal Cells" describes the differences at the level of senescence markers and differentiation between MSCs derived from synovial fluid of healthy horses and those with osteoarthritis. Although there are studies associating synovial MSC senescence with osteoarthritis, research on this topic is of interest because of its potential clinical implications for the treatment of this pathology.
Remarks
Q: It is surprising that since this is a study on senescence, no data on β-galactosidase activity have been presented. In my opinion, it would have been interesting to measure this activity in the two groups of MSCs studied to complete the results.
A: Thank you very much for your question. Indeed, it was one of the first experiment we performed, and the results did not match with the scientific literature regarding the β-galactosidase assay and cellular senescence. Based on our previous experience in MSCs, we were not surprised by the results. In MSCs isolated from different tissues in humans (adipose, bone marrow, umbilical cord, synovial membrane, dental pulp) and in some animal models (adipose tissue from sheep and synovial membrane and synovial fluid from horses), the presence of several lysosomes and active autophagic flux has been always observed, in agreement with some studies suggesting autophagy as a main condition for keeping stemness in stem cells (Garcia-Prat et al., 2016; doi: 10.1038/nature16187).The results we obtained (please, see the attached file) showed the lack of a significant difference between the healthy donors compared to the pathological ones. We think that in healthy donor the level of β-galactosidase activity is quite high due to the high number of lysosomes, confirmed also by TEM analysis. In general, our preliminary observation is that the number of lysosomes is significant high in MSCs isolated from donors (human and animals) in healthy conditions, while their number decrease in MSCs isolated from tissues involved in pathological conditions. Indeed, we have a similar result in vascular senescent MSCs isolated from a fragment of abdominal aorta aneurysm in which the β-galactosidase activity is like vascular MSCs isolated from healthy fragment of abdominal aorta (Teti et al., 2021 doi: 10.1016/j.mad.2021.111515).
Q: The design of the study is not entirely adequate. In each group the authors have introduced animals of very different ages ranging from 3 to 14 years old. As they themselves have shown in previous studies (Mazzotti et al., 2019 doi:10.3390/cells8101116), age influences the differentiation and viability of sf-MSCs in horses. Therefore, to the variability existing between different individuals in this type of studies, we must add the variability originated by the age difference. This makes it difficult to interpret the results. This may explain, for example, that animal No. 1 of the OA group, which is the youngest, presents very low values of P21 and P16 with respect to the older animals of the same group (Fig. 5) and less ROS production, with values very similar to those of the H group (Fig. 4).
A: This is a very important point and we are grateful for the question and the interesting discussion that will follow. We agree that in this kind of studies the variability between donors could be a problem amplified by age variability. Taking vantage from our previous experience (Mazzotti et al., 2019 doi:10.3390/cells8101116) in which morphological and functional characteristics of sf-MSCs isolated from healthy donors of different age were investigated, in this study we decided to collect samples from donors in a range from 3 to 14 years. Indeed, in our previous work we analyzed sf-MSCs isolated from donors with age ranging from 3 to 40 years (3Y, 12Y, 22Y, 40Y), demonstrating the appearance of gradual senescent features correlated with age. Taking into consideration that the most significant differences were visible starting from the group of 22 years and 40 years, in this study we decided to not overcome the range of 14 years old, trying to minimize the age variability as much as possible.
Regarding the signal of p21 protein in the n° 4 donor, we believe that this result is likely connected with a pre-senescent state, also confirmed by the slow DT and cell cycle analysis by flow cytometry but not by ROS production and BrdU assay. On the contrary, although p16 is considered a senescent marker, its expression is also tightly correlated with age (Diekman et al., 2018; doi: 10.1111/acel.12771). This could be the reason why we see a positive signal of p16 in healthy donor n° 2 and 4, corresponding to old donors in the pathological group.
Q: The statistics in Figure 6a and 6c should be revised or explained as it has been done. In both figures, according to the values represented, there is a large dispersion of data within each figure, making it difficult to understand that the differences are significant. In the case of Figure 6a, in the OA group there is only one animal with very high values with respect to the others. In the rest, it can be seen that the COL2A1 expression of No. 1 is the lowest of all, including those of group H, and that of animals 2 and 3 are at the average or lower than those of group H. Therefore, with these data it is very speculative to say that COL2A1 expression is higher in OA with respect to H. In my opinion, the study of chondrocyte marker gene expression shows no clear differences between the two groups of animals and should not be highlighted in the manuscript. Only with a larger number of donors of more homogeneous age, they could have more reliable data.
A: Thank you very much for having underlined this point. We had been expecting a low level of COL2A1 expression in all the OA donors, in agreement with our hypothesis that cellular senescence could impair the chondrogenic ability of OA sf-MSCs. Although the high variability between the different OA donors, the statistical analysis performed (Student’s t-test between OA sf-MSCs vs H sf-MSCs) demonstrated a significant difference compared the two groups. The entire experiment, starting from the seeding of cells to Real Time PCR, was performed three times, always obtaining the same results. We agree with you that just one OA donor showed a high expression of COL2A1 mRNA and deleting the corresponding data the differences between the two groups are not significant anymore. The interesting point of this result is that the n° 4 OA donor represents the one with the strongest senescent features, suggesting a correlation with an increase of COL2A1 expression. Our study is based on a low number of donors and represents just a preliminary result. In the future, we aim to increase the sample to better strength our hypothesis. We changed the paragraph 2.8 in agreement with your suggestion and the discussion in the section relative to chondrogenic differentiation (line from 428 to 443).
Q: Minor remarks:
A: We are grateful to you for the suggestions regarding the minor remarks and changed the manuscript following your points.
On line 281 it would be convenient to put "glycosaminoglycans (GAG)", due to the fact that GAG has not been defined before.
In the figure legends it is better to put the meaning of the statistical differences at the end of the legend and not to repeat the same text several times. For example, in the legend of Figure 5, line 260 repeats the same text as lines 265-266.

Round 2
Reviewer 4 Report
The authors have correctly answered my questions. In my opinion the article can now be published